# The Role of Autophagy in Pancreatic Cancer: From Bench to the Dark Bedside

**DOI:** 10.3390/cells9041063

**Published:** 2020-04-24

**Authors:** Kıvanç Görgülü, Kalliope N. Diakopoulos, Ezgi Kaya-Aksoy, Katrin J. Ciecielski, Jiaoyu Ai, Marina Lesina, Hana Algül

**Affiliations:** 1Comprehensive Cancer Center Munich, Technische Universität München, 81675 Munich, Germany; nina.diakopoulos@tum.de (K.N.D.); ezgi.kaya@tum.de (E.K.-A.); katrin.ciecielski@tum.de (K.J.C.); jiaoyuai@163.com (J.A.); marina.lesina@tum.de (M.L.); 2Department of Gastroenterology, The First Affiliated Hospital of Nanchang University, Nanchang 330006, China

**Keywords:** autophagy, pancreatic cancer, tumor microenvironment, therapy

## Abstract

Pancreatic cancer is one of the deadliest cancer types urgently requiring effective therapeutic strategies. Autophagy occurs in several compartments of pancreatic cancer tissue including cancer cells, cancer associated fibroblasts, and immune cells where it can be subjected to a multitude of stimulatory and inhibitory signals fine-tuning its activity. Therefore, the effects of autophagy on pancreatic carcinogenesis and progression differ in a stage and context dependent manner. In the initiation stage autophagy hinders development of preneoplastic lesions; in the progression stage however, autophagy promotes tumor growth. This double-edged action of autophagy makes it a hard therapeutic target. Indeed, autophagy inhibitors have not yet shown survival improvements in clinical trials, indicating a need for better evaluation of existing results and smarter targeting techniques. Clearly, the role of autophagy in pancreatic cancer is complex and many aspects have to be considered when moving from the bench to the bedside.

## 1. Introduction

Pancreatic cancer (PC) is one of most deadly cancer types mainly due to delayed diagnosis, high metastatic capacity, and aggressive local progression. In spite of therapeutic and diagnostic advances in other cancer types, clinical management of PC remains limited, crucially affecting patient health. Indeed, median survival of PC patients is merely 6–9 months and 5-year survival rate is as low as 9% [1].

The most prevalent type of PC is pancreatic ductal adenocarcinoma (PDAC), which harbors multiple genetic alterations. Most frequent genetic alterations promoting pancreatic tumorigenesis include *KRAS* activation, *TP53*, *CDKN2A*, and *SMAD4* mutation [2]. All of these genetic changes influence tumorigenesis in both a cell-autonomous and non-cell autonomous manner, affecting protein synthesis, cell growth, cell metabolism, and proliferation, collectively deranging cellular homeostasis. One of the most distinguished processes maintaining homeostasis is autophagy. Autophagy orchestrates a variety of tasks including retention of cellular functions, release of building blocks, and support of metabolic activities through engulfing intracellular substrates into double-membraned autophagosomes and routing them for lysosomal degradation [3].

As the recent advances go on, our understanding of autophagy continuously grows by the discoveries of several autophagy regulator proteins, substrates, and receptors in yeast and mammalian cells. Autophagy can be classified into three forms, namely macroautophagy, chaperone-mediated autophagy (CMA), and microautophagy [4]. At the same time, autophagy can be selective and specifically target various organelles, e.g., mitochondria, endoplasmic reticulum (ER), peroxisomes, nucleus, lysosome, or other cellular components like lipid droplets and aggregates, to sustain cellular homeostasis in normal physiology and under pathological conditions [4].

For the first time in 1981, Flaks et al. mentioned the existence of autophagy during pancreatic carcinogenesis. In their study, the researchers detected autophagy in Syrian hamsters after carcinogen-mediated PC induction [5]. This observation subsequently paved the way for the development of therapeutic approaches targeting autophagy. Now, the knowledge of how autophagy acts in PC is being expanded by studies independently analyzing macroautophagy, selective autophagy, and autophagy regulators.

The present review centers on the role of autophagy in PC in terms of tumor development, aggressiveness, and therapeutic vulnerabilities. Moreover, the importance of autophagy will be highlighted not only in the cancer core, but also in the tumor micro- and macroenvironment.

## 2. Autophagy and Molecular Mechanisms

Research into autophagy received a great amount of attention after the discovery of 15 autophagy-related (Atg) proteins in yeast in 1993 [6]. Since then, many scientists have focused on autophagy and its role in health and diseases, inspired also by the recent Nobel Prize in Physiology/Medicine in 2016.

The main function of autophagy is to degrade unrequired material, damaged intracellular compartments, and various aggregates, breaking them down into reusable cellular building blocks or metabolic substrates. In addition, autophagy takes up pathogens that invade cells and protects from infections. By being a gatekeeper of cell homeostasis, autophagy counteracts multiple types of diseases including cancer, which is usually based on defects in normal cellular function [7,8].

Different types of autophagy exist depending on the stimulus, the selectivity, and the machinery that is being utilized. In the following subsections, we will provide a brief overview, with a focus on the underlying molecular mechanisms. 

### 2.1. Execution of Autophagy

Depending on the molecular players involved, autophagy can be separated into macroautophagy, CMA, or microautophagy. The point of convergence for all these pathways is the lysosome, where the degradation process takes place [9].

Macroautophagy, commonly referred to as autophagy, involves the uptake of cellular material into double-membrane autophagosomes prior to delivery to lysosomes [10]. Macroautophagy is initiated by a multi-protein complex known as the ULK-complex (Unc 51-like kinase 1, FIP200, ATG101, ATG13), which acts as a stress and nutrient sensor inside the cell (see below). Assembly of the ULK-complex triggers multiple phosphorylation events collectively leading to nucleation of the phagophore, the precursor of mature autophagosomes, on the assembly sites at the ER. The occurring phosphorylation events target proteins on the class III PI3K complex I (PI3KC3) including Beclin 1-regulated autophagy protein 1 (AMBRA1), which under steady state conditions tethers PI3KC3 to components of the cytoskeleton keeping it in place and away from the ER [11]. PI3KC3 consist of the catalytic vacuolar protein sorting (VPS)34, Beclin-1, p115, and Atg14 and phosphorylation of multiple of these proteins release and activate PI3KC3. PI3KC3 in turn initiates local production of phosphatidylinositol-3-phosphate (PI3P) at the ER, recruiting PI3P effector proteins and their targets responsible for the elongation of the phagophore. The most important targets are two ubiquitin-like conjugation systems: (1) Atg7/Atg10, which conjugate Atg5 and Atg12, and (2) Atg5/Atg12, which binds to Atg16L1, is recruited to the nascent phagophore, and conjugates, with the help of Atg7/Atg3, phosphatidylethanolamine (PE) to GABARAP/light chain 3 (LC3) family of proteins. LC3 proteins are firstly cleaved by Atg4 cysteine proteases to form LC3 I, which after conjugation with PE forms LC3 II. LC3 II is the main component of the autophagosomal membrane and critical for selective autophagy (see below). After elongation, the autophagosome matures, taking up cellular constituents, and finally closes. Fusion with lysosomes into autolysosomes and degradation of the engulfed material follows along with nutrient recycling [10]. Of note, the existence of an alternative form of macroautophagy has been proven, which does not require Atg5/Atg7/LC3-modification, but rather relies on Rab9-dependent fusion of isolation membranes with vesicles derived from the Golgi apparatus or late endosomes [12,13]. Alternative macroautophagy is important in mitochondrial elimination during erythrocyte maturation, degradation of insulin granules in glucose-deprived beta-cells, and protection against inflammatory bowel disease [14,15,16].

Microautophagy describes the direct internalization of cytosolic constituents by lysosomes or late endosomes/ multivesicular bodies [17,18]. Distinct types have been described in mammalian cells depending on cargo selectivity, mechanism of internalization, and targeting vesicles [18,19]. The molecular machinery involved may include certain core Atg-proteins and/or Niemann-Pick type C (NPC) in the invagination step along with the endosomal sorting complex required for transport (ESCRT) proteins during vesicle fission/sealing. However, detailed description of microautophagy is still missing [20]. Recently, it has been shown that the cytosolic chaperone heat shock protein 70 (Hsc70) may bind to certain proteins and deliver them for degradation by endosomes/lysosomes, conferring more specificity to microautophagy [17].

CMA is a selective form of autophagy and starts with the binding of Hsc70 onto a sequence of five peptides (KFERQ) in the substrates [17,21]. Approximately 40% of the mammalian proteins contain this motif and can become targets of CMA if the motif is exposed [21,22]. After substrate recognition, Hsc70 binds to Lamp2a on lysosomes, which then multimerizes and initiates translocation of the unfolded protein into the lysosomal lumen. Translocation also requires a lysosomal-resident form of Hsc70, which does not exist in all the types of lysosomes, making some lysosomes specific for CMA. The importance of CMA is seen in its selectivity for certain protein targets. In cancer for example CMA exhibits a pro-tumorigenic role not only because of protein quality control but also because of the selective degradation of regulatory proteins (e.g., pyruvate kinase and p53 enhancing glycolytic rates/intermediates thus promoting tumor growth [23,24] or pro-apoptotic proteins [25] therefore promoting survival). In non-cancer cells however, CMA has a tumor-suppressive role and should be preserved/ restored [21].

In the following part, we will briefly describe ways to stimulate autophagy and mechanisms underlying cargo selectivity, with a focus on canonical macroautophagy (herein mentioned as autophagy).

### 2.2. Stimulation of Autophagy

Autophagy can either be baseline/constitutive or it can be stimulated. Constitutive autophagy occurs at a low level, degrading unfolded and damaged proteins, therefore providing constant cell protection and supply of metabolic substrates. In response to certain stimuli, autophagy can be highly activated leading to large-scale degradation of unspecific or specific substrates (see below). The best-described stimuli are nutrient shortage and intracellular damages [10]. Pathogens can also stimulate autophagy, however they will not be described in this review.

Lack of nutrients and specifically amino acids lead to dissociation of mammalian target of rapamycin (mTOR) complex 1 from the ULK complex, alleviating ULK inhibition and initiating the autophagy cascade. In addition, transcription factor EB (TFEB), a master regulator of lysosomal biogenesis, is negatively regulated by mTORC1 and activated upon starvation [10]. Low levels of ATP/AMP, and thus cellular energy, also activate autophagy by stimulating LKB1/AMPK leading to inhibition of mTORC1 and activation of ULK. Other levels of control can be exerted through transcriptional regulation, e.g., via the Forkhead box class O (FoxO) transcriptional factors [26] and bromodomain-containing protein 4 (BRD4) [10]. In addition, multiple core autophagy proteins are targets of regulation, e.g., Beclin-1, which is inhibited by Bcl-2, Akt, and EGFR or Vps34, which can be acetylated in mouse liver [10].

Cellular damage is the other determinant of autophagy activation. DNA damage [27], hypoxia, oxidative stress [28], etc., can stimulate autophagy through multiple effectors and transcription factors. Organelle damage and/or protein aggregation also triggers autophagy through specific pathways. Autophagy, therefore can be highly specific and solely degrade mitochondria, peroxisomes, ER, ferritin, liposomes, or the nucleus (see recent review Johansen T et al. [29]) to name just a few.

### 2.3. Cargo Selection

An important determinant of specific versus bulk autophagy is the molecules involved in cargo selection. Specific autophagy is usually triggered by organelle damage or protein aggregation, leading to targeted degradation (see types mentioned above). Selective autophagy receptors, which can be soluble and membrane-bound, interact with LC3-proteins at the inner autophagosomal membrane through their LC3-interaction region (LIR) [29]. Through other domains they recognize their cargo, recruiting it to the autophagosome. Cargo recognition can either be by direct interaction with the substrate or by binding to “eat me” signals like ubiquitin, which are placed on the substrate [29]. Many of the receptors can also di-/multimerize, forming “bodies”, which are discussed to act as platforms for phagophore assembly. The most prominent example and first discovered selective receptor is p62/SQSTM1, which recognizes among others ubiquitinated protein aggregates, mitochondria, and peroxisomes. NBR1, NDP52, OPTN, and TAX1BP1 are other examples, but the list is still growing. For a recent review on these receptors refer to Johansen et al. [29].

## 3. Autophagy and Pancreatic Cancer

### 3.1. The Role of Autophagy in Pancreatic Cancer

#### 3.1.1. Macroautophagy and Pancreatic Cancer

Metabolism and PC are tightly connected [30]. Indeed, prominent PC risk factors are among others dietary ingredients, diabetes mellitus type 2, and obesity [31]. Moreover, clinical presentation of PC commonly includes cachexia [32]. Autophagy is well known to critically drive cellular and systemic metabolism [33]. For this reason, the liaison between autophagy and PC was not totally unforeseen but remained to be proven.

In 1981, the first study showing the occurrence of autophagy during pancreatic carcinogenesis was published. Researchers chronically exposed Syrian hamsters to the carcinogen N-nitroso-bis(2-hydroxypropyl)amine (BHP), inducing formation of pseudoductular structures, which constitute precursors of pancreatic carcinogenesis arising from the acinar compartment [5]. During this de-differentiation of acinar cells, zymogen granules and granular ER were selectively targeted by autophagy [5]. In a following study, autophagic capacity of rat premalignant and malignant cells was shown to be higher than in a normal pancreas after azaserine-induced PDAC [34]. Moreover, cycloheximide, an inhibitor of autophagy, did not block the accumulation of autophagic vacuoles [34]. For the first time, it was clearly shown that autophagy was augmented in neoplastic cells compared to normal host tissue and not subject to standard regulation (cycloheximide block). Thereafter, the need for detecting the fine tuners of autophagy arose. Akar et al. showed that protein kinase C delta (PKC delta) determines tissue transglutaminase 2 (TG2)-expression, which hinders autophagy in pancreatic cancer cells (PCCs) [35]. TG2 siRNA subsequently induced autophagy not through mTOR but through Beclin-1 [35,36]. Therefore, the role of PKC delta in autophagy was mTOR independent but Beclin-1 dependent through TG2 [35]. This study also proved that the effect of autophagy regulators in cancer might differ among different cancer types.

As discussed earlier, LC3 is a dynamic marker of autophagy and has been used in many studies. Fujii et al. applied LC3 staining to tissues of resected PC patients and revealed a correlation between LC3 punctate pattern in the peripheral area of PC and poor patient outcome and shorter disease-free survival [37]. Additionally, autophagy occurred at different levels along the cancer tissue, especially in nerve and cancer cells. However, 9 out of 71 patients had no LC3 punctate pattern in cancer cells although these patients had strongly positive stained nerve cells as an internal control. Thus, therapeutic targeting of autophagy can be done albeit only after stratification of patients dependent on their autophagic status [37]. Following this, several autophagy related proteins including Atg5, Ambra1, Beclin-1, LC3, and Bif-1 were analyzed. Importantly, the expression levels of all proteins were significantly correlated. But the level of expression differed according to T-stage [38]. The translational meaning of autophagy related proteins during cancer progression was subsequently evoked.

In addition to survival related factors, inflammation related factors also influence autophagy. Stimulation of the induced receptor for advanced glycation end products (RAGE) drives NF-κB and MAP kinase signaling contributing to inflammation. Kang et al. showed that specific knockdown of RAGE in PCCs reduced cell survival via lowering autophagy levels [39]. RAGE was maintaining autophagy levels through decreased phosphorylation of mTOR and increased Beclin-1/VPS34-mediated autophagosome formation [39]. The same group observed release of the RAGE ligand, high mobility group protein 1 (HMGB1), during several tumor cell death types [39,40]. Furthermore, they depicted that PCCs utilize HMGB1 endogenously as a pro-autophagic protein augmenting cancer cell survival and preventing programmed apoptotic cell death via dislocation of Bcl-2 from Beclin-1 [41]. Cell endogenous and exogenous HMGB1 therefore has different meanings, i.e., autophagy versus necroptosis, and should be considered in the autocrine communication of cancer cells during cell death.

Progress in the field of autophagy and PC steered researchers towards new therapeutic avenues. Yang et al. showed that pharmacologic or genetic restraining of autophagy convey increased DNA damage and decreased mitochondrial oxidative phosphorylation rate, eventually eradicating pancreatic tumor growth in vitro and in vivo [42]. Chloroquine (CQ), a lysosomotropic agent that prevents lysosomal acidification and subsequent substrate degradation, is a late stage inhibitor of autophagy and has been considered a treatment option for extremely therapy resistant PC patients [42]. Therapy resistant PC is known to contain multiple cancer stem cells (CSCs), which can survive oxygen and nutritional deprivation. Rausch et al. depicted co-expression of autophagy, CSC, and hypoxia markers via immunohistochemistry in PDAC patients [43]. This finding was also confirmed by another group showing hypoxia-inducible factor-1 alpha (HIF1a) induced autophagy during transition of non-stem PCCs into pancreatic CSC-like cells expressing CD133 [44]. CQ decreased the amount of CSCs in PC tissue. But interestingly, this effect was due to the inhibition of CXCL12/CXCR4 signaling and not autophagy [45].

Therapeutic solutions for patients however, depend on the collection of mutations each individual cancer harbors. Pancreatic cancer contains *TP53* mutations, which can crosstalk with autophagic signaling. Researchers have been trying to generate compounds, which might recover the wild-type function of p53. CP-31398 and RITA, p53-reactivating molecules, could control autophagic flux via Thr172 AMPK phosphorylation and regulation of the SESN1-2/AMPK/mTOR axis in both wild type and mutant *p53* pancreatic cancer cell lines [46]. The interaction between *p53* and autophagy has also been studied in humanized genetically modified mouse models (GEMM) of PDAC. Rosenfeldt et al. showed that *p53* status matters to determine the road of pancreatic carcinogenesis. Pancreas specific deletion of *Atg5* or *Atg7* hindered cancer development in GEMMs harboring one oncogenic *Kras* allele, the most frequent mutation in PC [47]. However, autophagy deficient mice survived less compared to mice harboring only the *Kras* mutation, because of pancreatic degeneration and exocrine insufficiency [47]. Surprisingly, additional *Trp53* homozygous deletion hastened tumor development via increased glucose uptake, glycolysis, and pentose phosphate pathway [47]. In this setting, pharmacological intervention with CQ also increased cancer formation. Interestingly, another study using *Trp53* heterozygous and *Kras* mutant mice with concomitant *Atg5*-deletion reinstated the function of autophagy as a late stage tumor promoter [48,49]. Furthermore, pharmacological inhibition of autophagy with CQ reduced tumor growth in GEMMs with *Kras* and *Trp53* mutations as well as in PC patient derived xenografts [49]. Up to then, the in vivo findings supported clinical trials with autophagy inhibitors in PC patients. At the same time however, oncogenic Ras can also regulate autophagy and by extent the outcome of autophagy inhibition on cancer development. Morgan et al. analyzed the effect of oncogenic Ras on both autophagic flux and CQ sensitivity in different cancer cells [50]. According to their findings, mutational status of Ras in multiple cell types elicited distinct responses to CQ treatment. The effect of Ras on CQ sensitivity and autophagic flux differed between lung cancer cells and other cells including human skeletal muscle myoblasts and human embryonic kidney cells [50]. However, they did not include the effect of oncogenic Ras on PCCs. Interestingly, Görgülü et al. have shown that *Kras*-mutated murine PCCs with monoallelic deletion of *Atg5* were resistant to CQ treatment [51]. Furthermore, monoallelic deletion of *Atg5* in *Kras* mutation harboring PC GEMMs increased cancer aggressiveness by several autophagy independent and cell autonomous pathways including mitochondrial morphology and function, differences in intracellular Ca^2+^ flux, and increased activity of extracellular cathepsin D and L [51]. These recent in vivo studies focusing on the role of autophagy in GEMMs are summarized in Figure 1.

Blocking autophagy might not be the golden solution for PC. Manent et al. observed that inhibiting autophagy increases reactive oxygen species in Ras-driven epithelial tissue, subsequently enhancing cell proliferation via Jun kinase activation [53]. Confirmation came from Wang et al. showing that autophagy inhibition increases aggressiveness in H-Ras mutation harboring PCCs via inducing the p62(SQSTM1)/RelA pathway and stimulating Ras-dependent epithelial-mesenchymal transition [54]. Interrupted autophagic flux leads to the accumulation of autophagy substrates including p62/SQSTM1, a protein with a pivotal role in PC development and which might explain the aforementioned observations. Indeed, Todoric et al. described p62 accumulation following IKKα- deficiency, subsequently enhancing Nrf2/Mdm2 signaling and promoting PC progression [55].

*SMAD4*, another frequently mutated gene, serves as a pivotal player in TGF-B signaling. TGFB1 induced autophagy in PCCs acts disparately depending on *SMAD4* status. In SMAD4 expressing PDAC cells, TGFB1-induced autophagy can increase proliferation and decrease migration. In SMAD4 negative cells on the other hand, it activates MAPK/ERK signaling and acts in a reversed manner [56]. Moreover, in Ras mutation-harboring cancer cells, MAPK/ERK and autophagy pathways work together to pursue cancer cell survival [57]. Shutting down both RAF kinases, key onco-effectors of Kras, and autophagy E1 ligase ATG7 achieved the best therapeutic results in this setting [57]. 

The role of macroautophagy in PC is entangled depending on several factors including stage of carcinogenesis, mutational status, peripheral mediators (low nutrients, oxygen), and cellular context. Therefore, more investigations are needed to clarify the function of autophagy and autophagy regulators during PC development and progression and to clinically utilize the results. 

#### 3.1.2. Other Autophagic Machineries and Pancreatic Cancer

The autophagic toolbox of PC is not only restricted to macroautophagy. Other autophagic pathways including CMA, microautophagy, and the different types of selective autophagy are part of this toolbox. Accordingly, all autophagic processes have to be considered in order to grasp the translational meaning of autophagy. 

Macroautophagy and CMA are often simultaneously induced under stress [58]. Lysosomal associated membrane protein-2 (LAMP-2), the CMA regulator, has a pivotal role in maintaining homeostasis of pancreatic acinar cells [59]. Therefore, CMA might act multifunctionally in PC. Eps8, an actin dynamics controlling protein, has been linked to the initiation of solid cancers [60,61]. Welsch et al. have shown co-localization of Eps8 with the late endosomal/lysosomal compartment in metastatic pancreatic cancer cell lines [61]. Furthermore, increased levels of Eps8 affected lysosomal size and morphology. In explanation, the authors detected KFERQ-like motifs on Eps8, which are required for the interaction with Hsc-70/ LAMP-2 and the flow of CMA [61]. Hsc-70 is the constitutive isoform of Hsp-70. Hsp-70, a highly maintained core protein conducing protein folding and restraining protein aggregation [62], is overexpressed in PC [63]. Hyun et al. showed that Hsp-70 suppression with quercetin sensitized Panc-1 and MiaPaCa-2 PCCs to gemcitabine [63]. Thus, Hsp-70 is not only a CMA regulator but also affects survival of cancer cells. Another study showed that knockdown of optineurin, a strongly expressed protein in PC, was linked to increased ER-stress and CMA hampering colony formation in human PCCs. Proliferation was not affected [64]. However, more studies are required to completely comprehend the role of CMA in PC.

Under several stressors such as low nutrients, hypoxia, and DNA damage, defective mitochondria are selectively eliminated via mitophagy to maintain metabolic homeostasis [65]. After stressors occur, PINK1 is situated on the outer mitochondrial membrane (OMM), initiating Parkin recruitment. Following this step, components of OMM get ubiquitinated by Parkin. These ubiquitin chains are phosphorylated by PINK1 triggering the signal for autophagic process [66]. In addition to PINK1, BNIP3/NIX are also the main regulators of mitophagy [67]. Rapid growth of cancer causes disequilibrium between the cellular oxygen consumption rate and oxygen supply. Under hypoxic conditions BNIP3 is increased through the transcription factor HIF-1 [65]. Accordingly, oncogenic Kras mutations increased Bnip3l/Nix and loss of Nix attenuated pancreatic carcinogenesis in vivo [68]. However, BNIP3-silencing by hypermethylation in PC was associated with chemoresistance and decreased survival [69,70]. In both settings genetic alterations of mitophagy regulators affected mitophagy [68]. Along the same lines, systemic deletion of Parkin or Pink1 accelerated pancreatic carcinogenesis in *Kras* mutated GEMMs via mitochondrial iron accumulation [71]. Thus, mitophagy regulators might act differently depending on the cell type and the tumor stage. For better assessment of mitophagy, mitochondrial morphology, and oxidative stress in vivo, Wilson et al. generated a conditional MitoTimer reporter mouse line [72]. Tissue specific in vivo imaging of mitophagy will probably increase the understanding of mitophagy and the role of its regulators during pancreatic carcinogenesis.

Researchers identified ferritinophagy by the help of quantitative proteomics in PCCs. Nuclear receptor coactivator 4 (NCOA4), a selective cargo receptor, was highly localized in autophagosomes, orchestrating ferritinophagy, i.e., selective autophagy of ferritin and iron, thus increasing the bioavailability of intracellular iron and by extension damaging reactive iron species [73]. In pursuit of this study, another group showed that inhibition of NCOA4 suppressed ferritin degradation and decreased ferroptosis in PCCs [74]. Moreover, decreased levels of Atg5 and Atg7 attenuated erastin-induced ferroptosis reducing cellular ferrous iron levels and lipid peroxidation in PCCs and fibroblasts [74]. Ferritinophagy therefore enriched and complicated our understanding of the reciprocal interaction between autophagy and cell death.

Recently, the knowledge from cellular pathways regulating induction of autophagy in PC has increased significantly. Since the study of Perrera et al. depicting the role of MiT/TFE transcription factors and their involvement during autophagy-lysosome gene regulation in PC, researchers have put more attention into this field. Perrera et al. have shown that the autophagy-lysosome axis in PC is mainly controlled by substantive nuclear translocation of MiT/TFE factors by IPO8/IPO7 mediated transport [75]. Moreover, these factors sustain autolysosome-originated pools of amino acids. Therefore, the autophagy-lysosome network helps the metabolic adaptation of cancer cells [75]. Following this study, Sakamaki et al. have shown that BRD4 can repress autophagic flux and control the autophagy-lysosome gene network via binding to histone lysine methyltransferase G9a [76]. On the other hand, up-regulation of *MiT/TFE* genes in several cancers including PC enhanced RagD-mediated mTORC1 induction, eventuating increased cell proliferation and cancer growth [77]. Unarguably, this autophagy-lysosome network in PC should be the main focus of further studies.

### 3.2. The Role of Autophagy in the Micro/Macroenvironment of Pancreatic Cancer

#### 3.2.1. Autophagy and Cancer Associated Fibroblasts

Dense stroma is a prominent feature of PDAC and consists of cellular and non-cellular components such as fibroblasts, pancreatic stellate cells (PSC), immune cells, blood vessels, extracellular matrix proteins (ECMs), growth factors, and cytokines [78]. Among those PSCs are considered one of the major stromal cell types interacting with PCCs and contributing to cancer progression [79]. PSCs become activated upon stimulation by PCCs, subsequently undergoing transition from the quiescent to the myofibroblast-like state. Activated PSCs lose their lipid droplets containing vitamin A, increase expression of α-smooth muscle actin (α-SMA), and secrete large amounts of ECM molecules, growth factors, and cytokines [80]. These components are critical for desmoplastic stroma formation and have a profound impact on proliferation, invasion, and chemoresistance in PCCs [81].

Recently, research has focused on the underlying mechanisms of PSC-activation and their effect on pancreatic tumorigenesis. Specifically, autophagy activation in PSCs seems to be a critical step during transformation from the quiescent to the active state. A previous study showed that genetic inhibition of autophagy in PSCs leads to increased lipid droplet count and decreased αSMA expression and thus retention of the quiescent PSC-state. Similarly, autophagy inhibition upon CQ-treatment [82], coenzyme Q10 [83], and FTY720, a structural analogue of sphingosine-1-phosphate (S1P) [84], blocks PSC-activation by suppressing AMPK and activating mTOR. Consequently, changes in MMP/TIMP-ratio and decreased TGB-1 expression lead to ECM-degradation and fibrosis attenuation [85]. Autophagy seems to be required to meet the high-energy demands of activated PSCs, characterized by hyper-secretion and hyper-proliferation [86].

However, the importance of autophagy in PSCs, or generally of activated PSCs themselves, for PC prognosis is under debate. So far two contradicting studies have been published. On the one hand, autophagy in PSCs supported tumor growth, peritoneal dissemination, and liver metastasis in vivo. Genetic or chemical manipulations could block tumor progression [82]. On the other hand, another study reported autophagy inhibition as a mechanism increasing PDAC-resistance to gemcitabine treatment [87].

The key to the described contradictions probably lies within the interactions between all cells of the tumor microenvironment. Desmoplastic reaction produces a hypo-vascular, nutrient poor microenvironment inducing metabolic stress in cancer and stromal cells. For adaptation metabolic re-wiring is required. Within a tumor, cancer cells are also metabolically heterogeneous, making the picture even more complex. Crosstalk mechanisms are part of metabolic re-wiring and occur between all cell types of the tumor microenvironment. For instance, PDAC cells in hypoxic environments produce lactate, which can be used by normoxic cancer cells as fuel for proliferation [88,89]. Moreover, PCCs produce growth factors inducing metabolic reprogramming in PSCs. PSCs respond by secreting other growth factors and thus reciprocally re-programming PCCs [90]. Recently, changes in concentration of lactate and alanine have been documented by using imaging studies during pancreatic carcinogenesis [91]. Sousa et al. showed that tumor cells induce autophagy in PSCs leading to alanine secretion. Alanine is subsequently converted to pyruvate and used as a glucose/glutamine alternative for the tricarboxylic acid (TCA) cycle, which is especially important under low glucose conditions [92]. In support, PSC-conditioned medium increased oxygen consumption rate (OCR) but not glycolysis of PCCs. Among all PSC-metabolites secreted, only alanine was capable of these effects [92]. Further, PSC-derived alanine exhibited the biggest release/uptake rate among 14 amino acids, exceeding even lactate. Of note, alanine-derived carbon can also be used to produce lipids and non-essential amino acids [92].

In summary, autophagy is a major player in the PCC/PSC metabolic crosstalk dictating adaptation to the challenges of the tumor microenvironment, as summarized in Figure 2. We would need to understand the complex interactions occurring within the tumor microenvironment in order to predict the clinical outcome of autophagy manipulation.

#### 3.2.2. Autophagy and Immune Cells

Currently not much is known about the role of autophagy in the immune microenvironment of PC. However, it has been shown that two distinct processes should be considered: autophagy in cancer cells and autophagy in immune cells [93]. 

Autophagy in cancer cells can facilitate cancer immune evasion, vesicle-mediated immune suppression, regulate programmed cell death 1 ligand 1 (PD-L1) expression, and influence immunogenic cell death (Figure 3). Along these lines, immune evasion can be modulated by autophagy through major histocompatibility complex I (MHC-I) expression in cancer cells [93,94,95]. It seems to be context-dependent, e.g., on tumor stage and/or immune phenotype, whether autophagy leads to MHC-I degradation or expression. For example, the autophagy-mediated MHC-I reduction seen in melanoma cells can be reversed through the addition of interferon-γ (IFN-γ) [96]; a finding supported in non-small-cell lung carcinoma (NSCLC), where radiotherapy-induced autophagy and IFN-γ secretion stimulated MHC-I expression in cancer cells and increased cytotoxic T-lymphocyte (CTL) infiltration [97]. Furthermore, autophagy was shown to inhibit the formation of an immunologic synapse between melanoma and natural killer (NK) cells through degradation of gap-junctional connexin 43 [98], thus reducing sensitivity towards NK-cell mediated lysis. Moreover, autophagy inhibition was shown to increase sensitivity towards CTLs [99]. Tumor cell-released autophagosomes (TRAPs) were shown to facilitate immunosuppression through induction of M2-like macrophages characterized by increased expression of PD-L1 and IL-10. TRAP-induced macrophages subsequently inhibited T-cell proliferation and promoted tumor progression through PD-L1 [100].

While these findings suggest that autophagy inhibition would be an unequivocal success in immune cancer therapy, contradictions exist. Importantly, although homozygous deletion of *Atg5* inhibited tumor formation in a murine model of pancreatic carcinogenesis, heterozygous deletion or incomplete knockdown led to increased tumor formation and metastasis. Furthermore, the tumors that formed in *Atg5* heterozygous mice contained greater numbers of pro-tumorigenic M2-macrophages than control mice, and *Atg5* heterozygous primary tumor cells had up-regulated expression of cytokines that regulate macrophage chemoattraction and differentiation into M2 [51]. Also, autophagy in cancer cells modulates PD-L1 expression and seems to play an important role in the induction of immunogenic cell death, a type of apoptosis that stimulates the development of anticancer T-cell responses [93].

Autophagy in immune cells can influence normal T-cell function, macrophage polarization, and regulatory T-cell (T_reg_) immunosuppression (Figure 4). Regarding T-cell function, autophagy inhibition enhanced stable surface expression of MHC molecules, but it also modified the peptide pool displayed on MHC diminishing the presentation of immunodominant epitopes. As a result, autophagy positively affected tumor antigen presentation and to an extent T-cell immunity [93,101,102]. Furthermore, autophagy was shown to be crucial for the maintenance of mitochondrial homeostasis and the degradation of proapoptotic and antiproliferative proteins thus ensuring T-cell survival, function, and memory T-cell formation [93].

Whereas autophagy inhibition seems to be an ill-advised approach in the context of T-cell mediated anti-cancer immune reactions, the opposite holds true for macrophage and T_reg_-responses. Indeed, autophagy inhibition using CQ was shown to repolarize tumor-associated macrophages from M2 to the anti-tumor M1 phenotype and reduce tumor growth in a model of murine laryngeal cancer [103]. Furthermore, knockout of autophagy genes induced apoptosis of T_reg_-cells alleviating immune suppression and thus leading to higher percentages of tumor infiltrating CTLs and smaller tumors in a model of murine colon cancer [104].

Since the current publication landscape is sparse and pancreatic tumors are often characterized by an M2-macrophage and T_reg_-rich immune microenvironment [105], more studies have to be conducted focusing on the immune cell specific effects of autophagy inhibition.

#### 3.2.3. Autophagy and the Macroenvironment of Pancreatic Cancer

Autophagy does not occur only in the tumor and tumor microenvironment during tumor development and progression, but also influences the whole organism. A prominent example is cachexia characterized by severe weight loss and irreversible muscle wasting. Cachexia is one of the deadliest systemic effects of PC accounting for almost 30% of PC related deaths [106].

Multiple pathways can lead to muscle wasting. The FOXO-family of transcription factors is well-described in this setting [107]. For example, FoxO3 orchestrates muscle wasting, interestingly through stimulation of autophagic/lysosomal as well as proteasomal degradation [108]. The role of autophagy in atrophying myocytes has also been confirmed in cancer patients stratified for cachexia. Johns et al. showed that Beclin-1 and ATG5 in addition to SMAD-/inflammation-related pathways were significantly increased in muscles of cachectic versus non-cachectic patients [109]. In support, LC3B was also increasingly detected in skeletal muscles of cachectic cancer patients [110]. Of note, not only macroautophagy regulators but also mitophagy-effectors, such as BNIP3 and NIX/ BNIP3L were found elevated in cachectic muscles [110]. Autophagic flux promotes muscle wasting, a characteristic feature of cancer-associated cachexia, in lung cancer and colon cancer bearing mice [111]. It has been shown that human gastrointestinal cancer-associated cachexia causes autophagy induction via LC3-II, ATG5, and ATG7 expression in addition to mitochondrial changes and apoptotic induction in the skeletal muscle [112]. These findings point to the possible role of autophagy in PC-associated cachexia. However, knowledge about this link remains limited and needs more attention for the understanding of autophagy in cancer-associated cachexia in PC.

The stromal compartment of PC is a critical fuel source for the cancer cells themselves. Crosstalk between PSCs and PCCs was described previously. Along the same investigatory lines, autophagy was systemically inhibited by mosaic deletion of *Atg4B* and the importance of autophagy in the tumor versus host tissue was analyzed [52]. Indeed, orthotopic transplantation of autophagy competent or autophagy deficient *Kras*-mutation harboring PCCs into systemically autophagy competent or autophagy deficient mice showed different effects. While autophagy competent hosts negatively influenced growth of autophagy deficient cancer cells, autophagy deficient hosts were dampening cancer growth of autophagy deficient cancer cells only in the early stages. Cancer growth was sustained during the late stages in autophagy deficient hosts and tumors, mimicking the therapeutic observations. Still, cancer growth was slightly less compared to cancer growth of autophagy competent tumors in autophagy competent hosts [52] (Figure 1).

Targeting cancer growth through systemic manipulation of autophagy requires further investigations. We still don’t have a complete picture of the effects of autophagy locally and systemically.

## 4. Targeting Autophagy as a Therapeutic Approach for Pancreatic Cancer

To date, PC remains one of the most deadly and most difficult diseases to hospitalize and cure. However, discoveries in the field of autophagy and its functions in cancer, the microenvironment, and the macroenvironment opened new opportunities and treatment avenues for patients.

Before the generation of specific autophagy-inhibitors, researchers tried to manipulate autophagic flux via upstream mechanisms. Ozpolat et al. found that protein kinase C-delta (PKC delta) inhibits autophagy via TG2 induction, an indicator of the metastatic phenotype and worsened patient prognosis in PC. Blocking PKC delta/TG2-axis resulted in cell death with autophagy implicating PKC delta/TG2 as a new therapeutic target [36]. Further, the PKC delta inhibitor, Rottlerin, caused intrinsic and extrinsic apoptosis in PC. However these effects were not dependent on PKC delta, but on eukaryotic elongation factor-2 kinase (eEF-2K). Indeed, Rottlerin decreased expression and stimulated ubiquitin/proteasome-mediated degradation of eEF-2K [113]. Genomic instability and epigenetic alterations, e.g., DNA methylation and histone modification, are other characteristics of PC. Interestingly, HDAC inhibitors influence not only the epigenetic but also the autophagic machinery. Suberoylanilide hydroxamic acid (SAHA), one of the most promising therapeutic agent and HDAC inhibitors, induced cell death with autophagy through dampening the Akt/mTOR pathway and activating ER-stress response in cancer cells [114,115]. Moreover, SAHA increased gemcitabine sensitivity of PCCs [116]. Additional cell death caused by autophagy inducing agent, triptolide, eliminated metastatic PCCs through suppression of Akt/mTOR/p70S6K axis and elevation of ERK1/2 pathway [117].

An easy and practical therapeutic road to follow in PC is to repurpose clinically established drugs. Omeprazole for example decreased proliferation rate of PCCs in non-cytotoxic concentrations. Moreover, omeprazole reversed the bi-phasic effect of 5-fluorouracil and modulated the lysosomal transport pathway interfering with autophagy and resulted in programmed cell death [118]. In support, autophagy inhibition decreased proliferation of PCCs after treatment with 5-fluorouracil and gemcitabine [119].

The culmination of discoveries regarding the role of autophagy in PC models jumpstarted trials using hydroxychloroquine (HCQ), another clinically well-known agent, as a neoadjuvant therapy in advanced PDAC patients. However, the results did not satisfy the expectations. Daily administration of HCQ to previously treated metastatic PC patients showed inconsistent autophagy inhibition along with insignificant therapeutic effects in PDAC patients [120]. Addition of HCQ to the well-accepted frontline therapy of PC, namely gemcitabine and nab-paclitaxel (GA), decreased 1-year survival rate and median overall survival compared to the non-HCQ group (11.1 months vs. 12.1 months) in advanced PC patients. Although combination of HCQ and GA improved some of the treatment related adverse effects including neutropenia and anemia, others like fatigue, nausea, peripheral neuropathy, visual changes, and neuropsychiatric symptoms worsened [121]. Genomic analysis of patient samples depicted no significant correlation between patient outcome and *p53* mutation status after treatments. Unfortunately, therefore, findings from milestone studies showing the link between *p53* status and the role of autophagy in PC were not directly translated to clinical settings.

After the discouraging clinical results, promising observations arrived from two studies utilizing a combination of ERK and autophagy inhibitors. Collectively, inhibition of Kras signaling or its downstream effectors instigated autophagy thus protecting PCCs against cytotoxic effects [122,123]. Time- and dose-dependent MEK1/2 inhibition by trametinib in cancer cells induced the LKB1/AMPK/Ulk1 signaling axis and subsequently autophagy activation. Stable knockdown of Kras also increased autophagic flux, as did doxycycline removal from doxycycline-inducible *KrasG12D* harboring murine PDAC cells. Additionally, ERK inhibition enhanced PDAC dependency on autophagy probably by affecting lysosomal acidification, glycolysis, and mitochondrial biogenesis. Moreover, single therapy resistant, only CQ or only trametinib, preclinical models of tumors were highly sensitive to combination therapy. Of note, the data failed to reproduce findings related to the link between *p53* status and autophagy in PC, a fact also not supported by the clinical trials described above. Translation of these findings in one patient supported the antitumor effect of the trametinib and HCQ combination by showing decreased tumor burden, tumor marker CA19-9, and resolved metastatic liver lesions. It should be considered however that PDAC can recur and thus patients should be followed up on for longer. Moreover, confirmation of the therapeutic utility of this combinatorial treatment should be sought in a bigger patient cohort. The field of autophagy in PC is still waiting for ongoing clinical trials to reach successful and applicable results (Table 1).

## 5. Conclusions

The roles of autophagy and autophagic regulators differ during PC development and progression. Therefore, targeting autophagy as a therapeutic approach in patients has not yet reached the expectations of in vitro and in vivo studies. Not surprisingly, we still have a lot of questions to address before we can completely understand the function of autophagy in PC. The time point when autophagy is triggered during pancreatic carcinogenesis is unknown. Ultimately, instead of targeting autophagy in fully developed tumors, it could be targeted as a preventive approach in higher risk patients by understanding pancreatic cancer specific autophagy regulatory pathways. In addition to already established autophagy regulatory pathways, we should try to understand the oncogenic mutation driven specific regulatory pathways of autophagy. Chemotherapies are having a hard time reaching the cancer core due to the dense desmoplastic reaction. So far, none of the studies systemically inhibiting autophagy has addressed this problem or at least analyzed the tissue distribution of autophagy inhibitors after treatment. Before showing the effect of autophagy inhibition on cancer, we should therefore aim to better understand the effect of systemic autophagy inhibition. Studies on metastasis would also benefit from the complete understanding of systemic autophagic effects. A lot of in vitro and in vivo data already exist. The task is to make them more translational. Careful data mining combined with patient details and concept re-evaluation will help in making future therapeutic approaches more successful.

## Figures and Tables

**Figure 1 cells-09-01063-f001:**
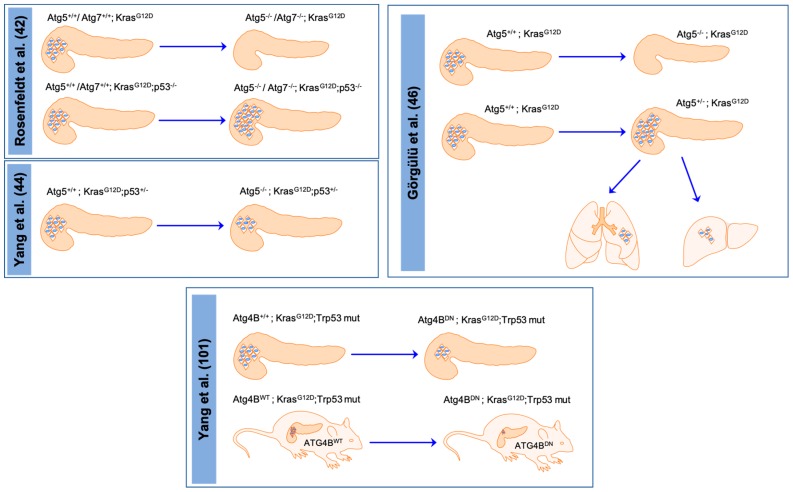
Recent in vivo studies showing the role of autophagy in genetically modified mouse models (GEMMs) of pancreatic cancer. Rosenfeldt et al. showed that homozygous deletion of *Atg7* or *Atg5* halts cancer development in the *Kras*-mutated murine PC model. However, the same genetic alterations increase cancer aggressiveness in the *Kras*-mutated *p53* deleted PC murine model [47]. At the same time, Yang et al. showed that autophagy acts as a tumor promoter in the *Kras*-mutated *p53* heterozygous murine PC model [49]. Following these studies, Görgülü et al. showed that monoallelic deletion of *Atg5* increased cancer incidence and metastatic dissemination in the *Kras*-mutated murine PC model. Later on, Yang et al. proved that compartment specific autophagy inhibition by *Atg4b* in cancer cells or the whole body suppresses cancer growth [52].

**Figure 2 cells-09-01063-f002:**
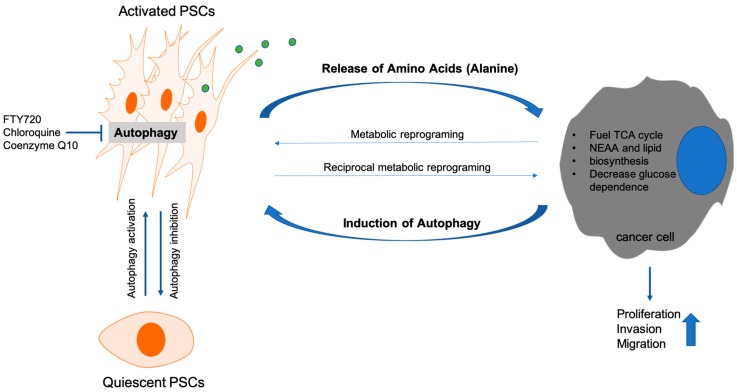
Intratumoral crosstalk between pancreatic stellate cells and pancreatic cancer cells. PSCs, pancreatic stellate cells; TCA, tricarboxylic acid cycle; NEAA, non-essential amino acids.

**Figure 3 cells-09-01063-f003:**
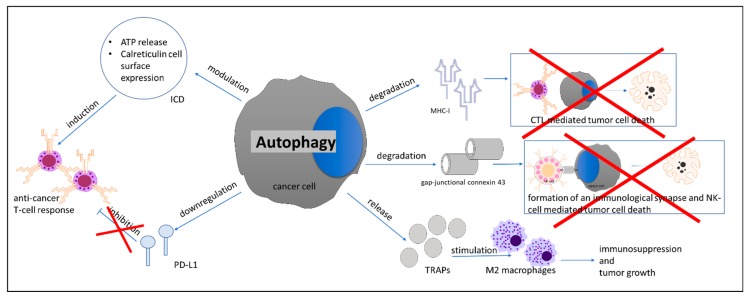
Effects of autophagy in cancer cells on the cancer immune microenvironment. ICD, immunogenic cell death; CTL, cytotoxic T lymphocyte; NK-cell, natural killer cell; TRAPs, tumor cell-released autophagosomes.

**Figure 4 cells-09-01063-f004:**
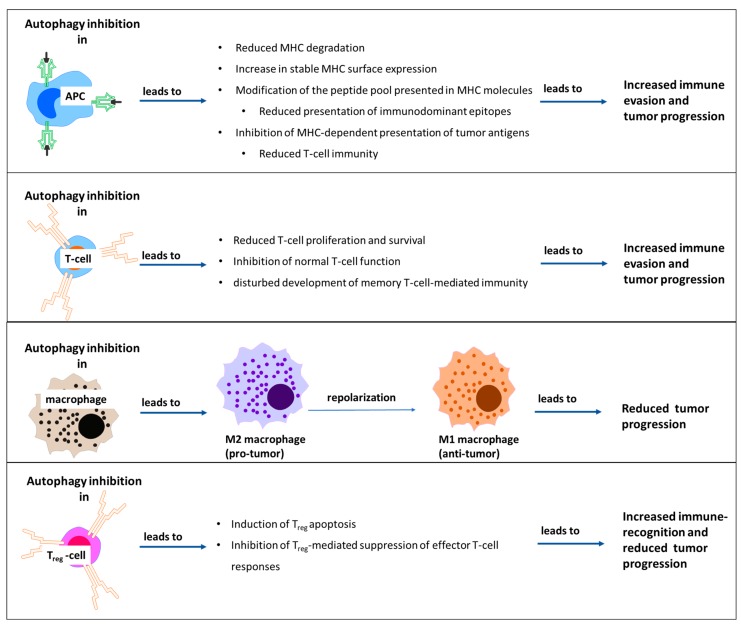
Effects of autophagy inhibition on cells of the immune system. APC, antigen-presenting cell; T_reg_, regulatory T-cell.

**Table 1 cells-09-01063-t001:** Clinical trials aiming to target autophagy in pancreatic cancer patients.

Trial ID	Clinical Trial	Treatment	Phase
NCT01777477	Adjuvant Effect of Chloroquine on Gemcitabine	Chloroquine + Gemcitabine	Phase 1
NCT01273805	Hydroxychloroquine in Previously Treated Patients With Metastatic Pancreatic Cancer	Hydroxychloroquine	Phase 2
NCT01494155	Short Course Radiation Therapy With Proton or Photon Beam Capecitabine and Hydroxychloroquine for Resectable Pancreatic Cancer	Capecitabine + Hydroxychloroquine + Proton or Photon Radiation Therapy	Phase 2
NCT04132505	Binimetinib and Hydroxychloroquine in Treating Patients With KRAS Mutant Metastatic Pancreatic Cancer	Binimetinib + Hydroxychloroquine	Phase 1
NCT03344172	Pre-Operative Trial (PGHA vs. PGH) for Resectable Pancreatic Cancer (17-134)	Gemcitabine + Nab-Paclitaxel + Hydroxychloroquine + Avelumab	Phase 2
NCT01978184	Randomized Phase II Trial of Pre-Operative Gemcitabine and Nab Paclitacel With or With Out Hydroxychloroquine	Gemcitabine + Abraxane + Hydroxychloroquine	Phase 2
NCT03825289	Trametinib and Hydroxychloroquine in Treating Patients With Pancreatic Cancer (THREAD)	Hydroxychloroquine + Trametinib	Phase 1
NCT01128296	Study of Pre-surgery Gemcitabine + Hydroxychloroquine (GcHc) in Stage IIb or III Adenocarcinoma of the Pancreas	Hydroxychloroquine + Gemcitabine	Phase 1/2
NCT01506973	A Phase I/II/Pharmacodynamic Study of Hydroxychloroquine in Combination With Gemcitabine/Abraxane to Inhibit Autophagy in Pancreatic Cancer	Hydroxychloroquine + Gemcitabine	Phase 1/2

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
