# Peer review of "The Role of Autophagy in Pancreatic Cancer: From Bench to the Dark Bedside"

_cells, 2020, doi:10.3390/cells9041063_

Round 1

Reviewer 1 Report

The manuscript by Görgülü and collaborators is a well written review about the different (and controversial) roles played by autophagy in cancer, with a focus on pancreatic cancer. There is a good review on the main mechanisms of autophagy, followed by a good description of autophagy on tumor cells and tumor stromal cells, with mentions to clinical trials and the use of autophagy inhibitors. The manuscript is well written, but with a few major and minor points that need to be addressed before publication.

Major points

Line 186: LC3 expression, or LC3 punctate pattern? The punctate pattern of LC3 is an indicator of autophagy, not solely expression. It is also not clear what type of correlation was found in the mentioned study. For instance, does Increased LC3 punctate pattern correlates to better outcome? Also, what does it mean “9 out of 71 patients had no LC3 expression”? IN what tissue? Only in the cancer cells or in the neighbor/non-cancer cells too? Please, describe better this part.

Line 324 and 331, page 8: Change “transformation” to “transition”, or any other similar word. Transformation, especially in a cancer review, is related to malignant transformation, which is not the case in this part.

Lines 426-438, page 11-12: How well established is that autophagy underlies the cachectic phenotype of PCC patients? This paragraph gives the impression that this phenomenon is well established, but references/details of this process are missing. Thus: If it is well established, please, cite appropriate references and describe the study(ies), providing the details available that support that. If it is just a hypothesis based on correlations, but no direct proof of concept experiments are available, please, make that clear.

Line 461, page 12: “autophagic cell death” is considered by some authors a misnomer (see, for instance: Kromer and Levine, Autophagic cell death: the story of a misnomer, Nat Rev Mol Cell Biol, 2009). What do authors mean by autophagic cell death? Death induced by autophagy, activation of autophagy during cell death (which could be either apoptosis or necrosis) such as a supportive role of autophagy in cell death via apoptosis, necrosis or other type of death?

Lines 516-518: What to authors mean by “Ultimately, instead of targeting autophagy in fully developed tumors it could already be targeted as a preventive approach in higher risk patients.”? Does it mean to inhibit autophagy in people that do not have developed tumor, but are considering at risk (because are carrier of cancer-associated mutations, dietary habits, etc.)? If this is the idea behind this phrase, it is already known that autophagy is tumor-suppressive, and such strategy could accelerate tumor development in people that do not have tumor.

Minor points

Italicize gene names. For instance, line 34 – page 1: TP53, CDKN2A should be TP53, CDKN2A.

Authors mention on line 110, page 3: “Approximately 40% of the mammalian proteins contain this motif and can become targets of CMA if the motif is exposed.” This is an interesting information, please, add appropriate reference at the end of this sentence.

Line 155-159, page 4: Since mitophagy regulated by Parkin/Pink1 dynamics is an important example of a selective form of autophagy that is implicated in cancer and other diseases, I believe authors could provide a brief detailed description of this mechanism in this section.

Lines 163-167: references are missing in this paragraph.

Line 244, page 6: the phrase “Interestingly, Görgülü et al have shown that monoallelic deletion of Atg5 in only Kras mutated murine PCCs reverse the cytotoxic effects of CQ” is somewhat confusing. Do authors mean that the murine PCCs have only Kras mutation? Or they are referring to murine PCC in which Kras is mutated (versus other PCC in which Kras is not mutated)?Please, reformulate this phrase.

In figure 1, please add a little more of description. Either in the legend, or in the figure itself. For instance, authors could add in the figure itself, on top of the arrows, something like: tumor growth, or tumor inhibition, to make it easier to understand the message behind it.

Line 380, page 9: define NSCLC.

In figure 4: Maybe authors could also include in this figure which of these pathways are pro or anti-tumor survival/proliferation. Similar to what was done in Figure 3.

Author Response

Response to Reviewer 1 Comments

Major Points

Point 1: Line 186: LC3 expression, or LC3 punctate pattern? The punctate pattern of LC3 is an indicator of autophagy, not solely expression. It is also not clear what type of correlation was found in the mentioned study. For instance, does Increased LC3 punctate pattern correlates to better outcome? Also, what does it mean “9 out of 71 patients had no LC3 expression”? IN what tissue? Only in the cancer cells or in the neighbor/non-cancer cells too? Please, describe better this part.

Response 1: Thank you very much for pointing this out. Unfortunately, this misleading terminology of LC3 has been used by the reference. In line 188, we have changed “LC3 expression” to “LC3 punctate pattern”. As, it is mentioned in the comment, accordingly we have included the type of this correlation correlation. To clarify this missing information, we have included “poor” and “shorter disease-free survival” in line 188 and 189 respectively. Authors of this cited study has used nerve cells as an internal positive control for each patient tissue. For this reason, cancer cells which did not give any positive staining for LC3 immunohistochemistry despite a positively stained nerve cells were judged as negative by authors.  We have briefly mentioned the origin of unstained area for LC3 which is cancer cells and their internal positive control by using LC3 punctate pattern in line 190 and 191 respectively.

Point 2: Line 324 and 331, page 8: Change “transformation” to “transition”, or any other similar word. Transformation, especially in a cancer review, is related to malignant transformation, which is not the case in this part.

Response 2: Thank you for this suggestion. You have raised an important point about the activation of pancreatic stellate cells over here. However, “transformation” has been used to mention this activation process in the consensus article of Erkan et al. Gut 2011- StellaTUM: current consensus and discussion on pancreatic stellate cell research. To prevent future confusions in this cancer review, we have changed “transformation” to “transition” in line 357.

Point 3: Lines 426-438, page 11-12: How well established is that autophagy underlies the cachectic phenotype of PCC patients? This paragraph gives the impression that this phenomenon is well established, but references/details of this process are missing. Thus: If it is well established, please, cite appropriate references and describe the study(ies), providing the details available that support that. If it is just a hypothesis based on correlations, but no direct proof of concept experiments are available, please, make that clear.

Response 3: Thank you very much for this comment. We totally understand your concern regarding the role of autophagy in pancreatic cancer associated cachexia. The knowledge about the role of autophagy in pancreatic cancer-associated cachexia is very limited. However, there are few studies showing the importance of autophagic pathways in cachexia related features. However, there is still no clear proof of concept experiment showing the role of autophagy in cachexia. We have updated this section with two references (111,112). Additionally, we have explained the need of new studies about the involvement of autophagy during pancreatic cancer-associated cachexia in line 480-481.

Point 4: Line 461, page 12: “autophagic cell death” is considered by some authors a misnomer (see, for instance: Kromer and Levine, Autophagic cell death: the story of a misnomer, Nat Rev Mol Cell Biol, 2009). What do authors mean by autophagic cell death? Death induced by autophagy, activation of autophagy during cell death (which could be either apoptosis or necrosis) such as a supportive role of autophagy in cell death via apoptosis, necrosis or other type of death?

Response 4: We agree with this comment. As Kroemer and Levine explained in this opinion article, “autophagic cell death” has been considered as a misnomer. Still after this opinion letter, many research articles kept using this terminology to represent type II Programmed Cell Death (see, for instance: Karch et al. Autophagic cell death is dependent on lysosomal membrane permeability through Bax and Bak, Elife 2017; Gen Leng et al. Activation of DRD5 (Dopamine Receptor D5) inhibits tumor growth by autophagic cell death, Autophagy 2017). When cell death comprises autophagy, it is determined as type II Programmed Cell Death or autophagic cell death. In included references, authors have observed that cell death was involving autophagic process during cell death with or without apoptotic cell death features. Therefore, accumulation of autophagic vacuoles and massive vacuole formation in the cytoplasm show that autophagic process may participate in cell death and as it is mentioned in the comment cell death may occur with autophagy. Therefore, we have changed “autophagic cell death” to “cell death with autophagy” in line 505,512, and 515. However, the scope of this review is into telling the role of autophagy in pancreatic cancer, instead of discriminating cell death types.

Point 5: Lines 516-518: What to authors mean by “Ultimately, instead of targeting autophagy in fully developed tumors it could already be targeted as a preventive approach in higher risk patients.”? Does it mean to inhibit autophagy in people that do not have developed tumor, but are considering at risk (because are carrier of cancer-associated mutations, dietary habits, etc.)? If this is the idea behind this phrase, it is already known that autophagy is tumor-suppressive, and such strategy could accelerate tumor development in people that do not have tumor.

Response 5: We agree with this comment. However, Kras-mutation harboring pancreatic cancer mouse models do not develop cancer and we clearly observe ADM and preneoplastic lesion formation after using pancreas specific deletion of autophagy regulator protein Atg5 (Görgülü et al. Gastroenterology 2019). This finding has been observed by others (Rosenfeldt et al. Nature, 2013). So basically, autophagy deficient mice do not develop Kras-mutation driven PDAC. It just enhances preneoplastic lesions in mice. For this reason, autophagy might be targeted as a preventive approach in higher risk patients carrying Kras-mutation by revealing pancreatic cancer specific autophagy regulatory pathways. We have updated our sentence in line 565 and 566. We also included one sentence in line 566 and 567 to emphasize the importance of pancreatic cancer specific autophagy regulatory pathways.

Minor Points

Point 1: Italicize gene names. For instance, line 34 – page 1: TP53, CDKN2A should be TP53, CDKN2A.

Response 1: We agree with this and have incorporated these changes throughout the manuscript.

Point 2: Authors mention on line 110, page 3: “Approximately 40% of the mammalian proteins contain this motif and can become targets of CMA if the motif is exposed.” This is an interesting information, please, add appropriate reference at the end of this sentence.

Response 2:  Thank you very much for this suggestion. We have added appropriate references at the end of this sentence in line 113.

Point 3: Line 155-159, page 4: Since mitophagy regulated by Parkin/Pink1 dynamics is an important example of a selective form of autophagy that is implicated in cancer and other diseases, I believe authors could provide a brief detailed description of this mechanism in this section.

Response 3: We appreciate your suggestion. However, in “Cargo Selection” part, we have already included a review from Johansen T et al. for detailed descriptions. But we thought that it is better to give this brief description aboutmitophagy controlled by Parkin/Pink1 in “Other Autophagic Machineries and Pancreatic Cancer” section. You can see included description in lines 310-313. Following that, line 313 and line 314 are updated.

Point 4: Lines 163-167: references are missing in this paragraph.

Response 4: Thank you very much for pointing this out. We have added missing references (30-33) in this paragraph(in line 165-169).

Point 5: Line 244, page 6: the phrase “Interestingly, Görgülü et al have shown that monoallelic deletion of Atg5 in only Kras mutated murine PCCs reverse the cytotoxic effects of CQ” is somewhat confusing. Do authors mean that the murine PCCs have only Kras mutation? Or they are referring to murine PCC in which Kras is mutated (versus other PCC in which Kras is not mutated)?Please, reformulate this phrase.

Response 5: Thank you for this suggestion. In our previous manuscript, we have shown that monoallelic deletion of Atg5 decreased sensitivity against to CQ in Kras-mutated murine PCCs. We have reformulated our phrase in line 250 and 251 to say where Kras-mutation harboring murine PCCs were sensitive to CQ, monoallelic deletion of Atg5 in these cells were resistant to CQ treatment.

Point 5: In figure 1, please add a little more of description. Either in the legend, or in the figure itself. For instance, authors could add in the figure itself, on top of the arrows, something like: tumor growth, or tumor inhibition, to make it easier to understand the message behind it.

Response 5: Thank you for this suggestion. We have briefly described Figure 1.  in the figure legend.

Point 6: Line 380, page 9: define NSCLC.

Response 6: Thank you for this point. We have defined NSCLC in line 415.

Point 7: In figure 4: Maybe authors could also include in this figure which of these pathways are pro or anti-tumor survival/proliferation. Similar to what was done in Figure 3.

Response 7: Thank you very much for pointing this out. As you mentioned, we have updated our figure by explaining immune evasion, tumor progression and immune recognition in each pathway.  

Reviewer 2 Report

Here Görgülü et al. review the role of autophagy in pancreatic cancer. They elucidate how autophagy impacts tumor progression at different stages of tumor development, how it influences tumor infiltrating T cells and the tumor microenvironment. Overall this is a good review of the role of autophagy in different areas of pancreatic cancer progression and its means to different immune cell subsets. The authors outlined the importance of additional research to be done and the necessity of correlating already acquired data to the autophagy process.

Some comments:

  • Please add a few sentences about increased lysosomal biogenesis in pancreatic cancer (PMID: 26168401).

  • Figure 4 is confusing. “Autophagy inhibition” has an arrow pointing down, which could mean a decrease in the inhibition of autophagy, or merely pointing to the cell. Then there are arrows pointing out with words like “reduction” or “inhibition” but it is not clear if the thing they are pointing to is reduced, or if autophagy inhibition causes a reduction in the reduction, hence an increase. There are triple negative statements here. Please re-work figure to simplify the message. Other figures could be similarly edited.

  • Line 13: “genetically most complicated” is vague terminology. Please rephrase or delete.

  • Line 388: replace “growth” with “progression” for clarity.

Author Response

Response to Reviewer 2 Comments

Point 1: Please add a few sentences about increased lysosomal biogenesis in pancreatic cancer (PMID: 26168401).

Response 2: We appreciate your suggestion. Therefore, we have updated our review with recent studies about autophagy-lysosome regulation and its transcriptional control in pancreatic cancer with three different references in line 337-349.

Point 2: Figure 4 is confusing. “Autophagy inhibition” has an arrow pointing down, which could mean a decrease in the inhibition of autophagy, or merely pointing to the cell. Then there are arrows pointing out with words like “reduction” or “inhibition” but it is not clear if the thing they are pointing to is reduced, or if autophagy inhibition causes a reduction in the reduction, hence an increase. There are triple negative statements here. Please re-work figure to simplify the message. Other figures could be similarly edited.

Response 2: We appreciate your suggestion. We updated Figure 4 with rearranging arrows and autophagy inhibition terms. Additionally, we have explained how each pathway acts in terms of immune evasion, tumor progression, and immune recognition. We have also changed other arrows pointing “reduction” and “inhibition” to arrows with “leads to” to make it more clear. Therefore, now it is more easier to follow this figure with the line of actions. Addition to Figure 4, we have also updated Figure 1 with the brief explanations of included studies in the figure legend.

Point 3: Line 13: “genetically most complicated” is vague terminology. Please rephrase or delete.

Response 3: We definitely aggree with this point. We have changed it to “most deadliest cancer types” in line 13.

Point 4: Line 388: replace “growth” with “progression” for clarity.

Response 4: Thank you very much for this suggestion. We changed “growth” to “progression” in line 424.
